# Comparison of Real-Time PCR and Droplet Digital PCR for the Quantitative Detection of *Lactiplantibacillus* *plantarum* subsp. *plantarum*

**DOI:** 10.3390/foods11091331

**Published:** 2022-05-03

**Authors:** Chang-Hun Choi, Eiseul Kim, Seung-Min Yang, Da-Som Kim, Seung-Man Suh, Ga-Young Lee, Hae-Yeong Kim

**Affiliations:** Resources and Department of Food Science and Biotechnology, Institute of Life Sciences, Kyung Hee University, Yongin 17104, Korea; legend1986@hanmail.net (C.-H.C.); eskim89@khu.ac.kr (E.K.); ysm9284@gmail.com (S.-M.Y.); dasove22@naver.com (D.-S.K.); teri2gogo@naver.com (S.-M.S.); gayoung.lee0731@gmail.com (G.-Y.L.)

**Keywords:** *Lactiplantibacillus plantarum*, real-time PCR, droplet digital PCR, detection, quantification, fermented food

## Abstract

Droplet digital polymerase chain reaction (ddPCR) is one of the newest and most promising tools providing absolute quantification of target DNA molecules. Despite its emerging applications in microorganisms, few studies reported its use for detecting lactic acid bacteria. This study evaluated the applicability of a ddPCR assay targeting molecular genes obtained from in silico analysis for detecting *Lactiplantibacillus* *plantarum* subsp. *plantarum*, a bacterium mainly used as a starter or responsible for fermentation in food. The performance characteristics of a ddPCR were compared to those of a quantitative real-time PCR (qPCR). To compare the linearity and sensitivity of a qPCR and ddPCR, the calibration curve for a qPCR and the regression curve for a ddPCR were obtained using genomic DNA [10^2^–10^8^ colony-forming units (CFU)/mL] extracted from a pure culture and spiked food sample. Both the qPCR and ddPCR assays exhibited good linearity with a high coefficient of determination in the pure culture and spiked food sample (*R*^2^ ≥ 0.996). The ddPCR showed a 10-fold lower limit of detection, suggesting that a ddPCR is more sensitive than a qPCR. However, a ddPCR has limitations in the absolute quantitation of high bacterial concentrations (>10^6^ CFU/mL). In conclusion, a ddPCR can be a reliable method for detecting and quantifying lactic acid bacteria in food.

## 1. Introduction

Lactic acid bacteria are involved in the spontaneous fermentations of foods such as meat, milk, fish, and vegetables [1]. They are commensal inhabitants of the human gastrointestinal tract and contribute to human health. As probiotics, lactic acid bacteria have shown that they may present beneficial effects, such as preventing diarrhea and inflammatory bowel disease [1,2,3]. The quantification of lactic acid bacteria is important for epidemiologic studies and their roles in various niche markets.

*Lactiplantibacillus plantarum* is a versatile species encountered in various niches, including meat, dairy, fish products, vegetables, and the human gastrointestinal tract [4,5]. *L. plantarum* is considered a safe probiotic, so it is widely used as an ingredient in fermented food and feed, such as cheese, milk, sauerkraut, and kimchi [6,7]. Between the two *Lactiplantibacillus plantarum* subspecies, *L. plantarum* subsp. *plantarum* provides a beneficial effect for the immune system, such as treating inflammatory diseases and mitigating pathogenic infections [8,9]. In a previous study, *L. plantarum* subsp. *plantarum* and *Lactiplantibacillus plantarum* subsp. *argentoratensis* affected the fermentation stage of vegetables, such as kimchi. *L. plantarum* subsp. *plantarum* was isolated only in fermented kimchi at a low temperature (4 °C), whereas *L. plantarum* subsp. *argentoratensis* was found only at a relatively high temperature (15 °C or 25 °C) [10]. Unlike *L. plantarum* subsp. *plantarum*, *L. plantarum* subsp. *argentoratensis* could not metabolize either methyl α-d-mannoside or melezitose [11].

A rapid method for the detection and quantification of *L. plantarum* subsp. *plantarum* in the food matrix is an essential tool for the food industry. However, prokaryotic systematics currently rely on labor- and time-consuming taxonomic approaches, including phenotypic characterization, variation analysis of 16S rRNA sequences, and DNA–DNA hybridization. Distinguishing closely related subspecies using these tools is difficult and often results in the misidentification of microorganisms [10]. Moreover, the molecular methods available for monitoring species or subspecies of *L. plantarum* are insufficient, as previous methods detect nontarget species of *Lactiplantibacillus* species or subspecies, including *L. plantarum* subsp. *argentoratensis*, *Lactiplantibacillus paraplantarum*, and *Lactiplantibacillus pentosus* [12,13]. Due to the limitations of these previous studies, there is an increasing demand for improving the current methods in studying prokaryotic systems [10].

The molecular-based detection and quantification of microorganisms have been successfully explored by real-time quantitative polymerase chain reaction (qPCR) in various food matrices [12,14]. A major advantage of a qPCR is that an amplification curve can be confirmed in hours instead of days, unlike conventional detection methods [15]. In addition, a qPCR is most commonly used as an efficient tool due to its high specificity and sensitivity [16]. A qPCR is a reliable and sensitive molecular tool applied and adopted in many different fields for the detection and quantification of genetically modified organisms (GMOs) [17,18], as well as for mutations and single nucleotide polymorphisms (SNPs) genotyping in the control of animal disease [19,20]. Recently, novel PCR-based methods for detecting and quantifying the molecular target have been introduced [21]. The droplet digital PCR (ddPCR) is a third-generation PCR tool [15]. A ddPCR mixture is divided into several partitions, each containing zero or at least one copy of the genomic DNA [15]. After amplification, partitions are counted as positive (presence of target gene) or negative (absence of target gene). The absolute quantification of the number of copies is performed using binomial Poisson statistics [15,22]. This allows a ddPCR to perform absolute quantification without using the calibration curve [15]. A ddPCR has been applied previously in various fields to quantify and detect genomic DNA targets, such as GMOs, viruses, pathogenic bacteria, antibiotic resistance genes, and vertebrate [23,24,25,26,27,28,29].

This study applied a ddPCR assay to detect and quantify *L. plantarum* subsp. *plantarum* in a food sample and compared the specificity and sensitivity of a qPCR targeting the *ydiC* gene. Moreover, the applicability of a ddPCR to detect lactic acid bacteria was discussed.

## 2. Materials and Methods

### 2.1. Bacterial Strains

*L. plantarum* subsp. *plantarum* and nontarget reference strains applied in this study were obtained from the Korean Collection for Type Cultures (KCTC, Daejeon, Korea), the Korean Culture Center of Microorganisms (KCCM, Seoul, Korea), the Korean Agricultural Culture Collection (KACC, Jeonju, Korea), the NITE Biological Resource Center (NBRC, Chiba, Japan), the Microorganism and Gene Bank (MGB, Gwangju, Korea), and the National Culture Collection for Pathogens (NCCP, Cheongju, Korea) (Table 1). All tested strains were grown in lactobacilli MRS (Difco Laboratories, Sparks, MD, USA) broth at 37 °C for 48 h. Genomic DNA of reference strains was extracted using DNeasy Blood & Tissue kit (Qiagen, Hilden, Germany). The genomic DNA concentration and quality were measured using a Maestrogen Nano spectrophotometer (Maestrogen, Las Vegas, NV, USA). Genomic DNA was stored at −20 °C before use as a template for qPCR and ddPCR [30].

### 2.2. Primer and Probe Design

The *ydiC* gene (accession no. EFK30629.1) discovered by a pangenome analysis [12] was used as a target for detecting *L. plantarum* subsp. *plantarum*. The primer/probe set for detecting the *ydiC* gene was designed using the Primer Designer program (Scientific and Education Software, Durham, NC, USA). The primer and probe were designed considering the guanine-cytosine (GC) content and length to ensure high amplification conditions. In silico specificity was performed using in silico PCR amplification software (http://insilico.ehu.es/PCR/ accessed on 8 March 2022) [31] with genome sequences obtained from the GenBank sequence database. The sequences of the primer and probe used for qPCR and ddPCR assays and the amplicon size of the target gene are listed in Table 2.

### 2.3. qPCR Assay

The qPCR mixture consisted of 20 ng DNA template, 500 nM of each primer, 250 nM probe, 10 µL TaqMan™ Fast Universal PCR Master Mix (Applied Biosystems, Foster City, CA, USA), and purified water to a final volume of 20 µL. The amplification reaction was performed with the 7500 Fast Real-time PCR System (Applied Biosystems). The reaction was run at 50, hold for 2 min, followed by 95 °C for 10 min, then 40 cycles consisting of 95 °C for 15 s and 60 °C for 1 min per cycle. The output data were analyzed using ABI 7500 Fast Software (Applied Biosystems, Foster City, CA, USA). The specificity of primer and probe set was confirmed using 105 reference strains (Table 1). A calibration curve was constructed using genomic DNA from *L. plantarum* subsp. *plantarum* KACC 11451 with different concentrations [10^2^–10^8^ colony-forming units (CFU)/mL], as reported previously [32]. The viable cell was counted by the plate count method. Briefly, serial dilutions of cultured strain were grown on MRS agar and counted after incubation at 37 °C for 48 h. Amplification was performed thrice to construct the calibration curve. The calibration curves were constructed by plotting the Ct value and the number of cells.

### 2.4. ddPCR Assay

ddPCR was performed using 10 µL of 2 × ddPCR Supermix for probe (Bio-Rad, Pleasanton, CA, USA), 500 nM of each primer, 250 nM probe, 20 ng DNA template, and distilled water to a total volume of 20 µL. ddPCR was used to make the droplet mixture using the QX200 droplet generator (Bio-Rad). The droplet mixture was transferred to a PCR reaction plate and amplified with the following conditions: denaturation of 95 °C for 10 min, followed by 40 cycles of a two-step thermal profile consisting of 95 °C for 15 s and 60 °C for 60 s. The PCR product was incubated at 98 °C for 10 min and cooled to 4 °C until the droplets were read. The PCR reaction plate was transferred to the QX200 droplet reader (Bio-Rad). The number of positive (high level of fluorescence) and negative (low and constant level of fluorescence) droplets obtained were analyzed using QuantaSoft software (Bio-Rad, Pleasanton, CA, USA) [33].

### 2.5. Artificially Contaminated Milk Sample

To compare the performance characteristics of qPCR and ddPCR, *L. plantarum* subsp. *plantarum* was used to artificially contaminate food samples. The food sample was obtained from a market and confirmed absent of *L. plantarum* subsp. *plantarum* by qPCR. To prepare the spiked food sample, the milk sample was spiked with a pure culture of *L. plantarum* subsp. *plantarum* at a concentration of 10^8^ CFU/mL [34,35] and mixed for 2 min using a homogenizer (Stomacher Circulator 400; Seward Ltd., London, UK). The number of bacteria was determined by the plate counting method according to a previous study [32]. Briefly, 0.1 mL of an appropriate dilution of bacteria was spread on MRS agar and incubation at 37 °C for 48 h for bacterial cell counting. An aliquot of 1 mL spiked food sample was transferred to a sterilized tube, used for genomic DNA extraction, and subjected to qPCR and ddPCR assays.

## 3. Results and Discussion

### 3.1. Specificity of Primer by In Silico PCR

Before performing a qPCR and ddPCR, the inclusivity and exclusivity of the new primer and probe set designed from the *ydiC* gene were tested by an in silico PCR assay. The *ydiC* gene is a novel genetic marker for detecting *L. plantarum* subsp. *plantarum* obtained from a pangenome analysis [12]. The sequence identity with the primer, in silico PCR result, and amplicon size obtained from the *ydiC* gene of 56 *L. plantarum* subsp. *plantarum* and 140 other species are represented in Appendix A. The primer and probe generated a positive reaction with all *L. plantarum* subsp. *plantarum* strains, whereas the remaining nontarget species or subspecies produced a negative reaction. The amplicon size of all *L. plantarum* subsp. *plantarum* was 150 bp.

### 3.2. Evaluation of the Specificity and Sensitivity by qPCR

The accuracy of a qPCR and ddPCR depends on the specificity of the sequence or primer used in the experiment [36]. Many studies have reported detecting *L. plantarum* using the 16S rRNA sequence or housekeeping genes (atpD, recA, and dnaK) as qPCR marker genes [36,37,38]. However, these genes reported to date have high sequence homologies among other species or subspecies, such as *L. plantarum* subsp. *argentoratensis* and *L. paraplantarum*, and require an additional procedure for identification that is time-consuming and costly. Therefore, this study designed the primer and probe using the *ydiC* gene discovered by a pangenome analysis [12].

A qPCR was performed using *L. plantarum* subsp. *plantarum* and 104 other reference strains to determine the amplification efficiency and specificity of the primer targeting the *ydiC* gene. The target DNA of *L. plantarum* subsp. *plantarum* was successfully amplified (Figure 1A), whereas no amplification was observed for the 104 other reference strains tested, indicating that the primer was specific for *L. plantarum* subsp. *plantarum* detection, consistent with a previous result using the same target gene [12]. The calibration curve was constructed using the genomic DNA of *L. plantarum* subsp. *plantarum* ranging from 10^8^ to 10^3^ CFU/mL. When the calibration curves of a qPCR have *R*^2^ ≥ 0.98 and a slope from −3.1 to −3.6, it can be regarded as a high-efficiency primer [39]. A calibration curve had an amplification efficiency of 88.364% (Figure 1B), suggesting that the primer showed a high efficiency in detecting *L. plantarum* subsp. *plantarum*.

Propidium monoazide (PMA) combined with a PCR appears to be a potential method for distinguishing between living and dead cells [40,41,42]. Several studies have reported the application of PMA treatments to quantify viable bacterial cells in foods by a qPCR and ddPCR [40,43]. In this study, because genomic DNA was quantified without a PMA treatment, there is a disadvantage that viable, but non-cultivable (VBNC) cells cannot be quantified.

### 3.3. Evaluation of Specificity and Sensitivity by ddPCR

A ddPCR has the potential to be a robust method for the quantification and detection of microorganisms in food [44,45]. This study investigated the potential of a ddPCR for detecting *L. plantarum* subsp. *plantarum*. Similar to the qPCR assay, the high specificity of a primer was observed in the ddPCR method. The genomic DNA from the 105 strains was detected by a ddPCR, and no droplets were observed for the 104 nontarget reference strains, including *Lactiplantibacillus* species closely related to *L. plantarum* subsp. *plantarum* (Figure 2A). Moreover, the ddPCR assay was sensitive with good linearity (slope = 0.9197, *R*^2^ = 0.996 for detecting the *ydiC* gene) ranging from 10^8^ to 10^2^ CFU/mL (Figure 2B). The limit of detection in the genomic DNA obtained from the pure culture was 10^2^ CFU/mL, with a quantification value of 0.4 ± 0.11 copies (mean ± standard deviation). However, the ddPCR analysis failed to quantify the DNA when the *L. plantarum* subsp. *plantarum* population was >10^6^ CFU/mL (Table 3). In the ddPCR, reaction saturation was reached with more than 20,000 positive droplets, making it impossible to quantify this concentration [35].

### 3.4. Comparison of Sensitivity and Linearity of qPCR and ddPCR Assays

To compare the sensitivity and reliability, serial dilutions (10^8^–10^2^ CFU/mL) of the genomic DNA extracted from the pure culture and spiked food sample were used to determine the limit of detection and the limit of quantification. The limit of detection and the limit of quantification were calculated according to the previous studies [39,46]. The limit of detection of the qPCR was determined as 10^3^ CFU/mL in both the pure culture and spiked food sample (Table 3). On the other hand, the limit of quantification was determined as 10^4^ CFU/mL (Table 3). The qPCR showed a good linearity range of 10^8^–10^3^ CFU/mL with a 0.999 coefficient of determination (*R*^2^) in the genomic DNA of the pure culture and spiked food sample (Figure 3A,B). In contrast to the qPCR, the quantitative detection range of the ddPCR was from 10^6^ to 10^2^ CFU/mL (Table 3). The ddPCR assay showed the genomic DNA of the pure culture and spiked food sample with a good linearity (*R*^2^ = 0.996 and 0.998; Figure 3C,D). The ddPCR assay exhibited the lowest limit of detection value (10^2^ CFU/mL) and limit of quantification value (10^3^ CFU/mL) compared to the qPCR, showing that the ddPCR sensitivity was ten times higher than the qPCR detection (Table 3), consistent with previous studies in which the ddPCR is 10-fold more sensitive than the qPCR [15,44]. However, ddPCR droplets are positively saturated at >10^6^ CFU/mL bacterial concentrations, making the Poisson distribution estimation invalid and resulting in a narrower dynamic range than the qPCR. The limitation of the ddPCR reported in previous studies was the quantification of the target DNA when bacterial concentrations exceeded 10^6^ CFU/mL [15,22,44]. In this study, samples with a high bacterial abundance (>10^6^ CFU/mL) were not quantified. To accurately quantify these samples, they must be diluted and run again on a ddPCR, thus increasing the time and cost required for the analysis [15]. Therefore, it is necessary to confirm that the amount of the target DNA is within the measurement range before performing the experiment.

This study evaluated the applicability of a ddPCR to detect *L. plantarum* subsp. *plantarum*. Unlike a qPCR, a ddPCR does not require a calibration curve. A ddPCR allowed quantifying the absolute number of the target DNA added to the mixture by partitioning the PCR reagents and using the Poisson algorithm [21,47]. This may reduce the bias introduced in the qPCR assay, as the calculated bacterial concentrations had to be constructed for the calibration curve. Moreover, the higher sensitivity of a ddPCR than of a qPCR has been reported previously [15,22,44]. This would be an advantage in food samples having PCR inhibitors or containing low copies of target molecules [27,28]. The difference in sensitivity between the assays was attributed to the higher resistance of the ddPCR to inhibitors occurring in food matrices [25,35]. In a ddPCR, the target DNA is distributed over thousands of droplets that constitute separate reaction compartments, leading to a higher tolerance to inhibitors [35,48]. Although more expensive and time-consuming than a qPCR, a ddPCR is currently a more reliable tool for detecting microorganisms, viruses, and GMOs [24,40,43,49]. Previous studies have demonstrated that a ddPCR is a suitable analytical tool for detecting foodborne pathogens in the food matrix [44,45,50]. Similar to foodborne pathogens, a ddPCR is a useful tool for detecting low-level *L. plantarum* subsp. *plantarum*. Therefore, a ddPCR can be a suitable analytical tool for the quantitative detection of lactic acid bacteria, which is important in probiotic products, dairy products, or fermented food.

## 4. Conclusions

To the best of our knowledge, this is the first report that demonstrated the ddPCR assay to quantify *L. plantarum* subsp. *plantarum*. The ddPCR observed a lower limit of detection than the qPCR, which could be advantageous in foods with a low number of target species. The ddPCR represents an innovation in the molecular world and is very useful, sensitive, and reliable for overcoming different limits for *L. plantarum* subsp. *plantarum* quantification. At the same time, it is not an instrument that is accessible and easy to use in any laboratories and industries, both for costs and for the type of analysis. In conclusion, this study can be used as preliminary data for a future robust assay optimization and validation. This method, which enables the quantification of *L. plantarum* subsp. *plantarum* in the food samples, can be a useful tool in the food industry to evaluate the quality of fermented food products.

## Figures and Tables

**Figure 1 foods-11-01331-f001:**
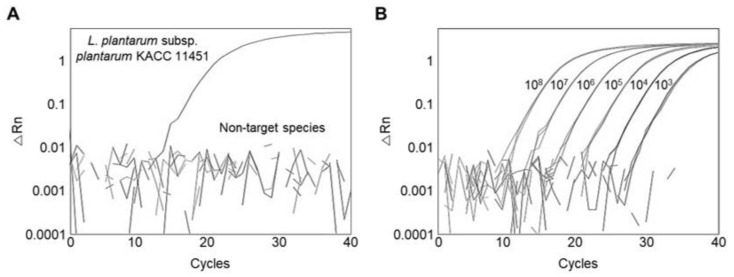
Assessment of primer specificity and sensitivity using qPCR. (**A**) Amplification plot of qPCR; (**B**) amplification plot of serial dilution of genomic DNA of *L. plantarum* subsp. *plantarum* KACC 11451.

**Figure 2 foods-11-01331-f002:**
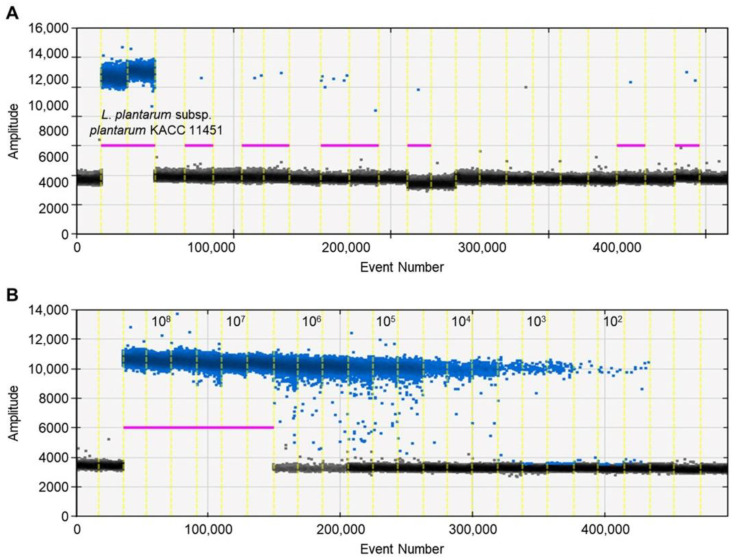
Assessment of primer specificity and sensitivity using ddPCR. The positive and negative droplets classified as the thresholds of individual wells are indicated in blue and black, respectively. The threshold is determined by the droplet reader and is shown as a horizontal line. (**A**) Specificity of primer by ddPCR; (**B**) quantification of genomic DNA of *L. plantarum* subsp. *plantarum* KACC 11451 by ddPCR.

**Figure 3 foods-11-01331-f003:**
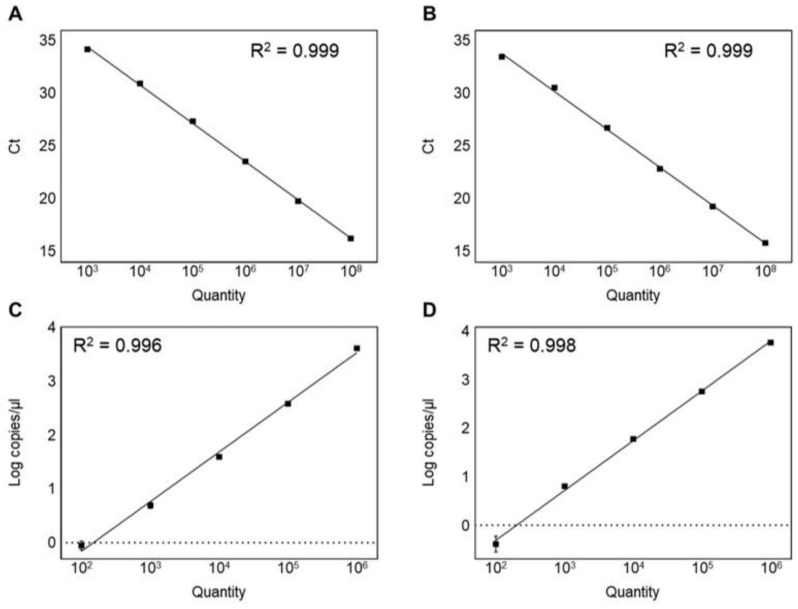
Linearity and sensitivity of two assays. (**A**) qPCR in analyzing genomic DNA of *L. plantarum* subsp. *plantarum* KACC 11451; (**B**) qPCR in analyzing spiked milk sample; (**C**) ddPCR in analyzing genomic DNA of *L. plantarum* subsp. *plantarum* KACC 11451; (**D**) ddPCR in analyzing spiked milk sample.

**Table 1 foods-11-01331-t001:** List of 105 reference strains.

Species	Strain No.
*Lactiplantibacillus plantarum* subsp. *plantarum*	KACC 11451
*Lactiplantibacillus plantarum* subsp. *argentoratensis*	KACC 12404
*Lactiplantibacillus paraplantarum*	KACC 12373
*Lactiplantibacillus pentosus*	KACC 12428
*Apilactobacillus kunkeei*	KACC 19371
*Bifidobacterium animalis* subsp. *animalis*	KCTC 3125
*Bifidobacterium animalis* subsp. *lactis*	KCTC 5854
*Bifidobacterium bifidum*	KCTC 3418
*Bifidobacterium bifidum*	KCTC 3440
*Bifidobacterium breve*	KACC 16639
*Bifidobacterium breve*	KCTC 3419
*Bifidobacterium longum* subsp. *infantis*	KCTC 3249
*Bifidobacterium longum* subsp. *longum*	KCCM 11953
*Companilactobacillus crustorum*	KACC 16344
*Companilactobacillus farciminis*	KACC 12423
*Companilactobacillus heilongjiangensis*	KACC 18741
*Enterococcus avium*	NCCP 10761
*Enterococcus avium*	KACC 10788
*Enterococcus casseliflavus*	KCTC 3638
*Enterococcus cecorum*	KACC 13884
*Enterococcus durans*	KCTC 13289
*Enterococcus durans*	KACC 10787
*Enterococcus faecalis*	KCTC 3206
*Enterococcus faecalis*	KACC 11859
*Enterococcus faecium*	KCTC 13225
*Enterococcus faecium*	KACC 11954
*Enterococcus faecium*	KACC 10782
*Enterococcus gilvus*	KACC 13847
*Enterococcus malodoratus*	KACC 13883
*Enterococcus mundtii*	KCTC 3630
*Enterococcus mundtii*	KACC 13824
*Enterococcus saccharolyticus*	KACC 10783
*Enterococcus thailandicus*	KCTC 13134
*Fructilactobacillus lindneri*	KACC 12445
*Lacticaseibacillus brantae*	KACC 17260
*Lacticaseibacillus camelliae*	KACC 17261
*Lacticaseibacillus casei*	KACC12413
*Lacticaseibacillus casei*	KCTC 13086
*Lacticaseibacillus casei*	KCTC 3110
*Lacticaseibacillus chiayiensis*	NBRC 112906
*Lacticaseibacillus manihotivorans*	KACC 12380
*Lacticaseibacillus pantheris*	KACC 12395
*Lacticaseibacillus paracasei* subsp. *paracasei*	KCTC 3165
*Lacticaseibacillus paracasei* subsp. *tolerans*	KCTC 3074
*Lacticaseibacillus rhamnosus*	KACC 11953
*Lacticaseibacillus rhamnosus*	KCTC 13088
*Lacticaseibacillus sharpeae*	KACC 11462
*Lactobacillus acetotolerans*	KACC 12447
*Lactobacillus acidophilus*	KACC 12419
*Lactobacillus acidophilus*	KCTC 3164
*Lactobacillus amylolyticus*	KACC 12374
*Lactobacillus amylophilus*	KACC 11430
*Lactobacillus amylovorus*	KACC 12435
*Lactobacillus brevis*	KCTC 3498
*Lactobacillus curvatus* subsp. *curvatus*	KACC 12415
*Lactobacillus delbrueckii* subsp. *bulgaricus*	KACC 12420
*Lactobacillus delbrueckii* subsp. *delbrueckii*	KACC 13439
*Lactobacillus delbrueckii* subsp. *lactis*	KACC 12417
*Lactobacillus gallinarum*	KACC 12370
*Lactobacillus gasseri*	KCTC 3163
*Lactobacillus gasseri*	KACC 12424
*Lactobacillus helveticus*	KACC 12418
*Lactobacillus jensenii*	KCTC 5194
*Lactobacillus johnsonii*	KCTC 3801
*Lactococcus lactis subsp. lactis*	KCTC 3769
*Lactococcus lactis subsp. lactis*	KCTC 2013
*Latilactobacillus sakei* subsp. *sakei*	KCTC 3603
*Lentilactobacillus buchneri*	KACC 12416
*Lentilactobacillus parabuchneri*	KACC 12363
*Leuconostoc carnosum*	KCTC 3525
*Leuconostoc citreum*	KCTC 3526
*Leuconostoc fallax*	KACC 12303
*Leuconostoc gelidum*	KACC 12256
*Leuconostoc gelidum* subsp. *aenigmaticum*	MGB 1000TE
*Leuconostoc gelidum* subsp. *gasicomitatum*	KACC 13854
*Leuconostoc gelidum* subsp. *gelidum*	KCTC 3527
*Leuconostoc holzapfelii*	KACC 17729
*Leuconostoc lactis*	KCTC 3528
*Leuconostoc mesenteroides* subsp. *dextranicum*	KACC 12315
*Leuconostoc mesenteroides* subsp. *mesenteroides*	KCTC 3505
*Leuconostoc pseudomesenteroides*	KACC 12304
*Levilactobacillus zymae*	KACC 16349
*Ligilactobacillus acidipiscis*	KACC 12394
*Ligilactobacillus agilis*	KACC 12433
*Ligilactobacillus ruminis*	KACC 12429
*Ligilactobacillus salivarius*	KCTC 3600
*Limosilactobacillus fermentum*	KACC 11441
*Limosilactobacillus fermentum*	KCTC 3112
*Limosilactobacillus fermentum*	KCTC 5049
*Limosilactobacillus mucosae*	KACC 12381
*Limosilactobacillus reuteri*	KCTC 3594
*Loigolactobacillus coryniformis* subsp. *coryniformis*	KACC 12411
*Streptococcus salivarius* subsp. *thermophilus*	KACC 11857
*Weissella beninensis*	KACC 18586
*Weissella cibaria*	KCTC 3746
*Weissella confusa*	KACC 11841
*Weissella halotolerans*	KACC 11843
*Weissella hellenica*	KACC 11842
*Weissella kandleri*	KACC 11844
*Weissella koreensis*	KACC 11853
*Weissella minor*	KCTC 3604
*Weissella paramesenteroides*	KACC 11847
*Weissella soli*	KACC 11848
*Weissella thailandensis*	KACC 11849
*Weissella viridescens*	KACC 11850

**Table 2 foods-11-01331-t002:** Information for primer and probe for detecting *L. plantarum* subsp. *plantarum*.

Name	Sequence (5′-3′)	Size (bp)
Plantarum_F	GGT GGC TGG TTG AGT GAT CT	150 bp
Plantarum_R	GCC GAT ACC GTT GGA AAT TA	
Plantarum_P	FAM-ACA GCT TGT TCT ACT AAC CGG CCT AGT CC-MGB	

**Table 3 foods-11-01331-t003:** Quantification of genomic DNA extracted from pure culture and spiked milk sample.

Conc. (CFU/mL)	Pure Culture ^1^		Spiked Food Sample	
	qPCR (Ct)	ddPCR (Copies)	qPCR (Ct)	ddPCR (Copies)
10^8^	16.37 ± 0.04 (9)	Saturated ^2^	16.13 ± 0.08 (9)	Saturated
10^7^	19.83 ± 0.03 (9)	Saturated	19.52 ± 0.05 (9)	Saturated
10^6^	23.37 ± 0.02 (9)	3287.78 ± 106.98 (9)	23.46 ± 0.07 (9)	4310 ± 295 (9)
10^5^	27.08 ± 0.15 (9)	318.44 ± 10.49 (9)	27.44 ± 0.12 (9)	427.22 ± 60.31 (9)
10^4^	30.67 ± 0.08 (9)	31.64 ± 1.9 (9)	31.25 ± 0.16 (9)	40.46 ± 6.58 (9)
10^3^	34.11 ± 0.31 (9)	3.42 ± 0.62 (9)	34.79 ± 0.3 (9)	3.98 ± 0.59 (9)
10^2^	ND ^3^	0.4 ± 0.11 (9)	ND	0.25 ± 0.14 (9)

^1^ Mean ± standard deviation (number of positive replicates on nine replicates analyzed). ^2^ DNA concentration at which the signal of the ddPCR assay was saturated. ^3^ ND, not detectable.

## Data Availability

Data is contained within the article or Appendix A.

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
