# Peer review of "Comparison of Real-Time PCR and Droplet Digital PCR for the Quantitative Detection of *Lactiplantibacillus* *plantarum* subsp. *plantarum"

_foods, 2022, doi:10.3390/foods11091331_

Round 1
Reviewer 1 Report
The authors used the genomic DNA of L. plantarum subsp. plantarum (ranging from 10^8 to 10^3 CFU/ml) as the standard curve.
However, CFU represent the live bacteria. Thus, using the PMA treatment is reasonably, please check the follow reference.
Scariot et al., 2018
Droplet digital PCR improves absolute quantification of viable lactic acid bacteria in faecal samples
Hansen et al., 2018
Absolute Enumeration of Probiotic Strains Lactobacillus acidophilus NCFM® and Bifidobacterium animalis subsp. lactis Bl-04® via Chip-Based Digital PCR
I think to use the total cell (dead cell+ live cell) as the standard curve is clearly, please check the follow reference.
Matsuki et al. (2004)
Quantitative PCR with 16S rRNA-gene-targeted species-specific primers for analysis of human intestinal bifidobacteria
Author Response
Response to Reviewer 1 Comments
The authors used the genomic DNA of L. plantarum subsp. plantarum (ranging from 10^8 to 10^3 CFU/ml) as the standard curve. However, CFU represent the live bacteria. Thus, using the PMA treatment is reasonably, please check the follow reference.
Scariot et al., 2018
Droplet digital PCR improves absolute quantification of viable lactic acid bacteria in faecal samples
Hansen et al., 2018
Absolute Enumeration of Probiotic Strains Lactobacillus acidophilus NCFM® and Bifidobacterium animalis subsp. lactis Bl-04® via Chip-Based Digital PCR
I think to use the total cell (dead cell+ live cell) as the standard curve is clearly, please check the follow reference.
Matsuki et al. (2004)
Quantitative PCR with 16S rRNA-gene-targeted species-specific primers for analysis of human intestinal bifidobacteria
Response: In this study, a calibration curve was constructed using genomic DNA from L. plantarum subsp. plantarum KACC 11451 with different concentrations (102–108 CFU/ml), as reported previously (Gómez-Rojo et al., 2015). The viable cell (102–108 CFU/ml) was counted by the plate count method. Also, in many studies, standard curve and regression curve were constructed by colony counting without PMA treatment (Wang et al., 2017). As you recommended, we added the sentence about the need for PMA treatment in the discussion section in lines 179-183 as follows:
Lines 179-183: Propidium monoazide (PMA) combined with PCR appears to be a potential method for distinguishing between living or dead cells [39–41]. Several studies have reported the application of PMA treatments to quantify viable bacterial cells in foods by qPCR and ddPCR [39,42]. In this study, since genomic DNA was quantified without PMA treatment, there is a disadvantage that viable but non-cultivable (VBNC) cells cannot be quantified.
Reviewer 2 Report
Original Submission
1.1. Recommendation: Minor Revision
2. Comments to authorsOverview and general recommendation:
Overall, the study is well designed, well performed and clearly presented. Although the flaws within the manuscript, I suggest its publication in case of minor revision.
Some indications for minor revisions are given below.
Line 37: "...in kimchi fermented...". It should be canged as follows: "...in fermented Kimchi...".
Line 90: Remove the second "The".
Try to improve the introduction with a brief section on lactic acid bacteria (LAB) and probiotics, in food and animal feed, presenting their importance and their benefactions twoards humans and animals.
The discussion section needs to be more developed. Try to add some recent references related to the same field.
Try to develop the conclusion with future outlook.
Check the punctuation all through the manuscript to make the meaning clearer.
Check the italic mode through the text, even in the reference list.
English language and stye need to be verified.
Author Response
Response to Reviewer 2 Comments
Overview and general recommendation:
Overall, the study is well designed, well performed and clearly presented. Although the flaws within the manuscript, I suggest its publication in case of minor revision. Some indications for minor revisions are given below.
Line 37: "...in kimchi fermented...". It should be canged as follows: "...in fermented Kimchi...".
Response: As you recommended, we revised the sentence in line 45 as follows:
Line 45: only in fermented Kimchi at low
Line 90: Remove the second "The".
Response: As you recommended, we removed the second “The” in line 99 as follows:
Line 99: The ydiC gene (accession no. EFK30629.1) discovered
Try to improve the introduction with a brief section on lactic acid bacteria (LAB) and probiotics, in food and animal feed, presenting their importance and their benefactions twoards humans and animals.
Response: As you recommended, we added the sentence in lines 31-36 as follows:
Lines 31-36: Lactic acid bacteria are involved in spontaneous fermentations of foods such as meat, milk, fish, and vegetables [1]. They are commensal inhabitants of the human gastrointestinal tract and contribute to human health. As probiotics, lactic acid bacteria have shown that they may present beneficial effects, such as preventing diarrhea and inflammatory bowel disease [1–3]. The quantification of lactic acid bacteria is important for epidemiologic studies and their roles in various niche markets.
The discussion section needs to be more developed. Try to add some recent references related to the same field.
Response: As you recommended, we added more supporting references and sentence in lines 179-183, 200-201, 248, and 253 as follows:
Lines 179-183: Propidium monoazide (PMA) combined with PCR appears to be a potential method for distinguishing between living or dead cells [39–41]. Several studies have reported the application of PMA treatments to quantify viable bacterial cells in foods by qPCR and ddPCR [39,42]. In this study, since genomic DNA was quantified without PMA treatment, there is a disadvantage that viable but non-cultivable (VBNC) cells cannot be quantified.
Lines 200-201: In the ddPCR, reaction saturation was reached more than 20,000 positive droplets, making it impossible to quantify this concentration [34].
Line 248: PCR inhibitors or containing low copies of target molecules [27,28].
Line 253: detecting microorganisms, virus, and GMO [24,39,42,48].
Try to develop the conclusion with future outlook.
Response: As you recommended, we added the sentence in lines 263-267 as follows:
Lines 263-267: ddPCR represents an innovation in molecular world, very useful, sensitive and reliable overcoming different limits for L. plantarum subsp. plantarum quantification. At the same time, it is not an instrument accessible and easy to use to any laboratories and industries, both for costs and for the type of analysis. In conclusion, this study can be used as preliminary data for a future robust assay optimization and validation.
Check the punctuation all through the manuscript to make the meaning clearer.
Response: As you recommended, we checked the punctuation all through the manuscript.
Check the italic mode through the text, even in the reference list.
Response: As you recommended, we checked the italic mode through the text.
English language and style need to be verified.
Response: The original manuscript was edited professionally by Enago Editing Services (reference number: INQ-8151396522). If the English level of the entire revised manuscript does not meet the level required by the journal, we will re-edit it to a native speaker.
Reviewer 3 Report
Review Report Foods “Comparison of real-time PCR and droplet digital PCR for the quantitative detection of Lactiplantibacillus plantarum subsp. plantarum”
Brief summary:
This article focuses on comparison of real-time PCR and droplet digital PCR for the quantitative detection of Lactiplantibacillus plantarum subsp. plantarum in pure culture and spiked milk matrix. L. plantarum is a bacterium used as fermentation starter and as probiotic in food industries and it can be also naturally found in different food and feed such as cheese, sourdough, yogurt but also meat products in addition to human gastrointestinal tract. The aim of the project is interesting to deal in depth, because it present a strategic molecular mean and assay for food chain monitoring in official control laboratories and in food industries.
Broad comments:
This article focuses on comparison of real-time PCR and droplet digital PCR for the quantitative detection of Lactiplantibacillus plantarum subsp. plantarum in pure culture and spiked milk matrix. L. plantarum is a bacterium used as fermentation starter and as probiotic in food industries and it can be also naturally found in different food and feed such as cheese, sourdough, yogurt but also meat products in addition to human gastrointestinal tract. The aim of the project is interesting to deal in depth, because it present a strategic molecular mean and assay for food chain monitoring in official control laboratories and in food industries. Anyway, the design assay is not too robust, in particular for that concerns the limit of detection/quantification (LOD/LOQ), both for qPCR and ddPCR. I suppose that the target gene ydiC is a multi-copy gene, because the sensitivity of 103 CFU/ml for pure culture and for spiking milk samples is not very low, considering that the assay should be adopted and applied also to complex food matrices, where matrix effect has to be considered (cheese derived products, meat derived products, vegetables…). Considering the qPCR linearity (log10= 3.3 Cq) it is strange that the amplification signal is lost from 103 to 102. So, I’m attending an explanation of this point of weakness and an experimental improvement, if possible. Indeed, if possible, I suggest to estimate LOD/LOQ in a rigorous way, testing more dilution in more replicates, following the guidelines reported below. Robustness parameters should be considered in these type of studies.
“European Commission. EUR 29015 EN. Verification of Analytical Methods for GMO Testing when Implementing Interlaboratory Validated Methods; Version 2. European Network of GMO Laboratories (ENGL). JRC Scientific and Technical Reports Luxembourg; European Commission: Brussels, Belgium, 2017.
Broeders, S.; Huber, I.; Grohmann, L.; Berben, G.; Taverniers, I.; Mazzara, M.; Roosens, N.; Morisset, D. Guidelines for validation of qualitative real-time PCR methods. Trends Food Sci. Technol. 2014, 37, 115–126.”
ddPCR represents an innovation in molecular world, very useful, sensitive and reliable overcoming different limits for targets quantification. At the same time it is not an instrument accessible and easy to use to any laboratories and industries, both for costs and for the type of analysis. In conclusion, does this report want to represent a preliminary study prior to a future robust assay optimization and validation?”
Specific comments:
- Introduction
The introduction has to be implemented in particular by emphasizing the large application of real-time PCR in different fields, integrating some studies and references. I suggest this kind of general sentence and relative references: “qPCR is a reliable and sensitive molecular tool applied and adopted in many different fields as for detection and quantification of GMO (Pierboni et al. 2016; Fraiture et al., 2021), as well as for mutations –SNPs- genotyping in control of animal disease (Lefever et al. 2019; Torricelli et al. 2021).
Line 67: other important and recent references dealing with ddPCR lack in the main text.
- Material and Methods
Line 90: please, remove “the” because is reported twice.
Line 95: please specify the software for in silico analysis (BLAST tool or other…?)
Line 133: Why was a concentration of 108 CFU/ml chosen for preparing spiked milk? Are there some supporting references?
Line 134: please detailed this important plate counting method
- Results and discussion
Lines 140-141: this sentence is not relevant, in my opinion.
Lines 188-189: what could be the explanation of this data? Please explain and contextualize this concept.
Line 207: please correct 108 into 106
Table 3: please correct spiking food sample into spiking milk sample
Table S1: the table has to be integrated with clearer results of negativity and positivity in exclusivity/inclusivity panel
I suggest to add more supporting references in this “discussion section” and to implement Conclusions section.
Presentation of the manuscript is good and clear as well as English language and style.
References
Better check the reported references in accordance with the format required by “Foods-MDPI” Journal

Author Response
Response to Reviewer 3 Comments
Review Report Foods “Comparison of real-time PCR and droplet digital PCR for the quantitative detection of Lactiplantibacillus plantarum subsp. plantarum”
Brief summary:
This article focuses on comparison of real-time PCR and droplet digital PCR for the quantitative detection of Lactiplantibacillus plantarum subsp. plantarum in pure culture and spiked milk matrix. L. plantarum is a bacterium used as fermentation starter and as probiotic in food industries and it can be also naturally found in different food and feed such as cheese, sourdough, yogurt but also meat products in addition to human gastrointestinal tract. The aim of the project is interesting to deal in depth, because it present a strategic molecular mean and assay for food chain monitoring in official control laboratories and in food industries.
Broad comments:
This article focuses on comparison of real-time PCR and droplet digital PCR for the quantitative detection of Lactiplantibacillus plantarum subsp. plantarum in pure culture and spiked milk matrix. L. plantarum is a bacterium used as fermentation starter and as probiotic in food industries and it can be also naturally found in different food and feed such as cheese, sourdough, yogurt but also meat products in addition to human gastrointestinal tract. The aim of the project is interesting to deal in depth, because it present a strategic molecular mean and assay for food chain monitoring in official control laboratories and in food industries.
Anyway, the design assay is not too robust, in particular for that concerns the limit of detection/quantification (LOD/LOQ), both for qPCR and ddPCR. I suppose that the target gene ydiC is a multi-copy gene, because the sensitivity of 103 CFU/ml for pure culture and for spiking milk samples is not very low, considering that the assay should be adopted and applied also to complex food matrices, where matrix effect has to be considered (cheese derived products, meat derived products, vegetables…). Considering the qPCR linearity (log10= 3.3 Cq) it is strange that the amplification signal is lost from 103 to 102. So, I’m attending an explanation of this point of weakness and an experimental improvement, if possible. Indeed, if possible, I suggest to estimate LOD/LOQ in a rigorous way, testing more dilution in more replicates, following the guidelines reported below. Robustness parameters should be considered in these type of studies.
“European Commission. EUR 29015 EN. Verification of Analytical Methods for GMO Testing when Implementing Interlaboratory Validated Methods; Version 2. European Network of GMO Laboratories (ENGL). JRC Scientific and Technical Reports Luxembourg; European Commission: Brussels, Belgium, 2017.
Broeders, S.; Huber, I.; Grohmann, L.; Berben, G.; Taverniers, I.; Mazzara, M.; Roosens, N.; Morisset, D. Guidelines for validation of qualitative real-time PCR methods. Trends Food Sci. Technol. 2014, 37, 115–126.”
Response: The target gene ydiC is a single copy gene. As you recommended, we determined the LOD and LOQ of qPCR and ddPCR according to the references and added sentence in lines 213-219 and 223-227 as follows:
Lines 213-219: To compare the sensitivity and reliability, serial dilutions (108–102 CFU/ml) of genomic DNA extracted from pure culture and spiking food sample were used to determine the limit of detection and the limit of quantification. The limit of detection and the limit of quantification was calculated according to the previous studies [38,45]. The limit of detection of qPCR was determined as 103 CFU/ml in both pure culture and spiking food sample (Table 3). On the other hand, the limit of quantification was determined as 104 CFU/ml (Table 3).
Lines 223-227: The ddPCR assay exhibited the lowest limit of detection value (102 CFU/ml) and limit of quantification value (103 CFU/ml) compared to qPCR, showing that ddPCR sensitivity was ten times higher than qPCR detection (Table 3), consistent with previous studies that ddPCR is 10-fold more sensitive than qPCR
Table 3: We added the LOD and LOQ results of qPCR and ddPCR to Table 3.
ddPCR represents an innovation in molecular world, very useful, sensitive and reliable overcoming different limits for targets quantification. At the same time it is not an instrument accessible and easy to use to any laboratories and industries, both for costs and for the type of analysis. In conclusion, does this report want to represent a preliminary study prior to a future robust assay optimization and validation?”
Response: This report wants to represent a preliminary study prior to a future robust assay optimization and validation. As you recommended, we added the sentence in lines 263-267 as follows:
Lines 263-267: ddPCR represents an innovation in molecular world, very useful, sensitive and reliable overcoming different limits for L. plantarum subsp. plantarum quantification. At the same time, it is not an instrument accessible and easy to use to any laboratories and industries, both for costs and for the type of analysis. In conclusion, this study can be used as preliminary data for a future robust assay optimization and validation.
Specific comments:
- Introduction
The introduction has to be implemented in particular by emphasizing the large application of real-time PCR in different fields, integrating some studies and references. I suggest this kind of general sentence and relative references: “qPCR is a reliable and sensitive molecular tool applied and adopted in many different fields as for detection and quantification of GMO (Pierboni et al. 2016; Fraiture et al., 2021), as well as for mutations –SNPs- genotyping in control of animal disease (Lefever et al. 2019; Torricelli et al. 2021).
Response: As you recommended, we added the sentence and reference in lines 66-69 as follows:
Lines 66-69: qPCR is a reliable and sensitive molecular tool applied and adopted in many different fields as for detection and quantification of genetically modified organisms (GMO) [17,18], as well as for mutations and single nucleotide polymorphisms (SNPs) genotyping in control of animal disease [19,20].
Line 67: other important and recent references dealing with ddPCR lack in the main text.
Response: As you recommended, we added important and recent references dealing with ddPCR in lines 77-78 as follows:
Lines 77-78: such as GMO, viruses, pathogenic bacteria, antibiotic resistance genes, and vertebrate [23–29].
- Material and Methods
Line 90: please, remove “the” because is reported twice.
Response: As you recommended, we removed the second “The” in line 99 as follows:
Line 99: The ydiC gene (accession no. EFK30629.1) discovered
Line 95: please specify the software for in silico analysis (BLAST tool or other…?)
Response: As you recommended, we added the software for in silico analysis in line 104 as follows:
Line 104: using in silico PCR amplification software (http://insilico.ehu.es/PCR/)
Line 133: Why was a concentration of 108 CFU/ml chosen for preparing spiked milk? Are there some supporting references?
Response: In many previous studies, a concentration of 108 CFU/ml chosen for preparing spiked food samples (Lei et al., 2020; Cristiano et al., 2021). As you recommended, we added supporting references in line 141 as follows:
Line 141: concentration of 108 CFU/ml [33,34]
Line 134: please detailed this important plate counting method
Response: As you recommended, we added the sentence in lines 142-145 as follows:
Lines 142-145: The number of bacteria was determined by the plate counting method according to a previous study [31]. Briefly, 0.1 ml of an appropriate dilution of bacteria was spread on MRS agar and incubation at 37°C for 48 h for bacterial cell counting.
- Results and discussion
Lines 140-141: this sentence is not relevant, in my opinion.
Response: As you recommended, we deleted this sentence.
Lines 188-189: what could be the explanation of this data? Please explain and contextualize this concept.
Response: As you recommended, we added the explanation of this data in lines 200-201 as follows:
Lines 200-201: In the ddPCR, reaction saturation was reached more than 20,000 positive droplets, making it impossible to quantify this concentration [34].
Line 207: please correct 108 into 106
Response: As you recommended, we corrected 108 into 106 in line 221 as follows:
Line 221: ddPCR was from 106 to 102
Table 3: please correct spiking food sample into spiking milk sample
Response: As you recommended, we corrected spiking food sample into spiking milk sample.
Line 208: Quantification of genomic DNA extracted from pure culture and spiking milk sample
Table S1: the table has to be integrated with clearer results of negativity and positivity in exclusivity/inclusivity panel
Response: As you recommended, we added a column for exclusivity/inclusivity and marked the results for negatively and positively in Table S1.
I suggest to add more supporting references in this “discussion section” and to implement Conclusions section.
Response: As you recommended, we added more supporting references and sentences in lines 179-183, 200-201, 248, 253, and 263-267 as follows:
Lines 179-183: Propidium monoazide (PMA) combined with PCR appears to be a potential method for distinguishing between living or dead cells [39–41]. Several studies have reported the application of PMA treatments to quantify viable bacterial cells in foods by qPCR and ddPCR [39,42]. In this study, since genomic DNA was quantified without PMA treatment, there is a disadvantage that viable but non-cultivable (VBNC) cells cannot be quantified.
Lines 200-201: In the ddPCR, reaction saturation was reached more than 20,000 positive droplets, making it impossible to quantify this concentration [34].
Line 248: PCR inhibitors or containing low copies of target molecules [27,28].
Line 253: detecting microorganisms, virus, and GMO [24,39,42,48].
Lines 263-267: ddPCR represents an innovation in molecular world, very useful, sensitive and reliable overcoming different limits for L. plantarum subsp. plantarum quantification. At the same time, it is not an instrument accessible and easy to use to any laboratories and industries, both for costs and for the type of analysis. In conclusion, this study can be used as preliminary data for a future robust assay optimization and validation.
Presentation of the manuscript is good and clear as well as English language and style.
Response: We very thanks to your critical reviews.
References
Better check the reported references in accordance with the format required by “Foods-MDPI” Journal
Response: As you recommended, we checked the reported references in accordance with the format required by “Foods-MDPI” Journal.
Round 2
Reviewer 3 Report
Dear Authors,
the entire work is interisting and the manuscript was Iintegrated and improved also accordingly to suggestions and comments. In my opinion, manuscript is publishable, but in proof preparation I recommend to insert amplification plot also in supplementary material in particular for LOD/LOQ computation and analysis. Please, verify some little english mistakes (for example in line 145: "incubation" has to be changed into "was incubated"). In Table S1 please convert "positively/negatively" into "positive/negative".
Kind regards.